# The Biological Function and Roles in Phytohormone Signaling of the F-Box Protein in Plants

**Keheng Xu [1], Nan Wu [1], Wenbo Yao [1], Xiaowei Li [1,*], Yonggang Zhou [2,*] and Haiyan Li [1,2,*]**

1. Engineering Research Center of the Chinese Ministry of Education for Bioreactor and Pharmaceutical Development, College of Life Sciences, Jilin Agricultural University, Changchun 130118, China; xh312319@163.com (K.X.); wunan026@163.com (N.W.); yaowenbo0205@163.com (W.Y.)
2. Sanya Nanfan Research Institute, College of Tropical Crops, Hainan University, Haikou 570228, China
* Correspondence: xiaoweili1206@163.com (X.L.); yonggang0408@163.com (Y.Z.); hyli99@163.com (H.L.); Tel.: +86-0898-66150179 (Y.Z. & H.L.)

**Abstract:** The ubiquitin–proteasome pathway (UPP) is an important protein degradation pathway that can participate in the regulation of the physiological process of organisms by specifically removing abnormal peptides and degrading cell regulators. UPP mainly involves three enzymes, among which the E3 ubiquitin ligase function is central to UPP. E3 ubiquitin ligases can recruit substrate protein for ubiquitination, and they have various forms. Among them, the Skp1–Cul1–F-box (SCF) complex is the most representative member of the cullin RING ubiquitin ligases type in RING-domain E3 ligases, being mainly composed of Cullin 1, Skp1, Rbx1, and F-box proteins. The F-box protein is the key component for SCF to perform specific functions. The F-box protein is one of the largest protein families in plants, and its family members are involved in the regulation of many key physiological processes, such as growth and development of plants and the response to external stimuli. Herein, we briefly review the structure, classification, function, and hormone signaling pathways of F-box proteins.

**Keywords:** ubiquitin–proteasome pathway; SCF complex; F-box protein; biotic and abiotic stresses; phytohormone signaling

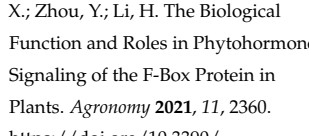

## 1. Introduction

Protein degradation is an important post-transcriptional regulation process that plays an extremely important role in cell physiological activities. The ubiquitin–proteasome pathway (UPP) is an important protein degradation pathway. In yeast and animals, the UPP can remove most abnormal polypeptides, degrade cell regulators, and regulate many physiological processes [1]. It is thus believed that this pathway also plays a similar role in plants [2]. For instance, mutations in the UPP component can block embryogenesis and hormone response in Arabidopsis. It is also estimated that 80% of proteins in organisms will undergo ubiquitin–proteasome-mediated degradation. Therefore, the UPP has attracted great attention [3,4]. The UPP mainly involves three enzymes: ubiquitin-activating enzyme E1, ubiquitin-conjugating enzyme E2, and ubiquitin ligase E3. Members of E3 are mainly classified into three subfamilies: RING-domain E3 ligases (RING E3s), homologous to E6AP C-terminus E3s (HECT E3s), and RING-in-between-RING E3s (RBR E3s) [5]. RING E3s are the largest subfamily of E3s and the type of Skp1-Cul1-F-box (SCF) E3 ligases is the most representative member of Cullin-RING ubiquitin ligases in RING E3s [6]. The SCF complex contains four subunits, namely, the S-phase kinase-associated protein (Skp1), the scaffold protein Cullin1 (CUL1), the RING-box1 (Rbx1), and the F-box protein. Skp1, CUL1, and Rbx1 are invariant and can form a core scaffold through interactions, while the F-box protein is the flexible subunit in the SCF complex [6,7].

Kumar and Paietta in 1995 first discovered the F-box protein when they studied the WD-repeat sequence [8]. It was not until the following year that the F-box proteins

were named when Bai et al. discovered that the F-box motif was a motif essential for protein–protein interaction as they studied the cyclin F protein [9]. With the development of technology, an increasing number of *F-box* genes have been discovered in the following decades. Interestingly, the quantitative difference between F-box members in different species is countless (Figure 1). By comparison, the number of *F-box* genes in plants is far more than other eukaryotes. The model plants *Arabidopsis thaliana* and *Medicago sativa* contain 694 and 972 *F-box* genes, respectively [10–12]. Important crops such as rice, wheat, soybean, corn, and chickpea contain 687, 409, 509, 359, and 285 *F-box* genes, respectively [13–17]. Economic crops such as upland cotton, apple, and pear contain 592, 517, and 226 *F-box* genes, respectively [7,18,19]. On the other hand, there are 69 *F-box* genes in humans and 22 in fruit flies [20]. With so many *F-box* genes in plants, the *F-box* gene family recognizes many different ubiquitin–proteasome system (UPS) degradation targets, which indicates that they can participate in the regulation of many biological processes. At present, studies have shown that *F-box* genes can regulate plant growth and development, including hormone, root development, self-incompatibility, senescence, and response to abiotic and biotic stress [21–25]. However, there are few comprehensive reviews of the role of *F-box* genes in plant growth and development, including its role in stress response. Moreover, the understanding of the metabolic pathways involved in the *F-box* gene in plants unclear. Here, we summarized the function and metabolic pathways of the *F-box* gene family in plants so as to lay the foundation for further study of the *F-box* genes.

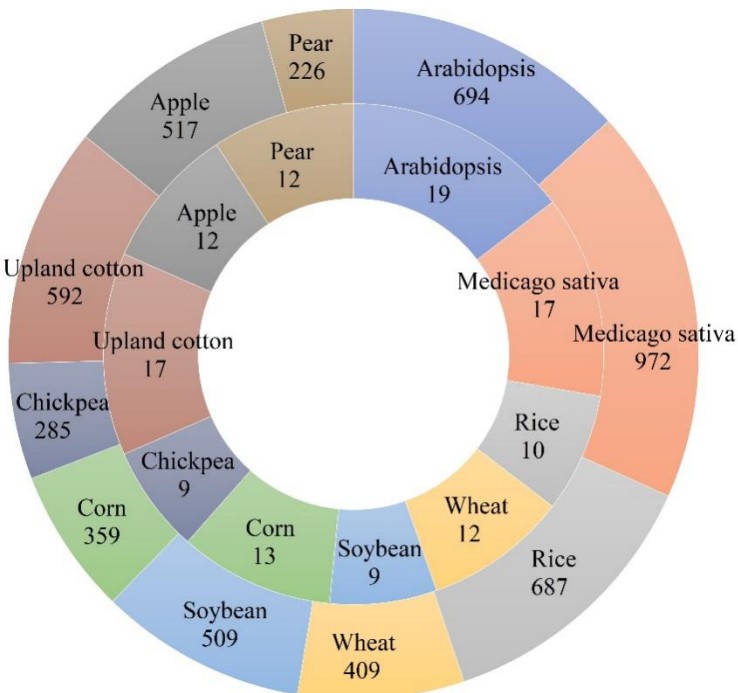

**Figure 1.** The number of plant F-box genes and subfamilies in different plants. The outer loop indicates the number of F-box genes in different plants and the inner loop indicates the number of F-box subfamilies in different plants.

## 2. The Structure and Action Processes of the F-Box Protein

F-box proteins are a class of proteins that widely exist in eukaryotes with F-box motifs. The F-box motif contains 40–50 amino acids, mainly located at the N-terminal, which can interact with the Skp1 protein [20]. The C-terminal of the F-box protein usually contains a (or several) secondary structure that mediates substrate-specific recognition, which can recruit substrate protein for ubiquitination. In addition, the C-terminal domain is the basis of F-box protein classification [1] (Figure 2a). For the convenience of research, researchers divided the enormous F-box family into different subfamilies according to its C-terminal domain. Therefore, the F-box protein family in animals was divided into three

subfamilies, namely, the FBXL, which represents leucine-rich repeats at the C-terminal; FBXW, which represents WD40 repeats at the C-terminal; and FBXO, which is a protein with other secondary structures or unknown domains at its C-terminal [26,27]. However, because their families are too large and the number of different species is different in plants, the number of subfamilies is also different. In general, studies generally include nine subfamilies, namely, the FBU, whose C-terminal contains the unknown structure; FBL, whose C-terminal contains leucine-rich repeats; FBK, whose C-terminal contains the Kelch domain; FBA, whose C-terminal contains an F-box associated domain; FBD, whose C-terminal also contains an F-box domain; FBT, whose C-terminal contains the tubby domain; FBP, whose C-terminal contains the phloem protein-2 domain; FBW, whose C-terminal contains the WD40 domain; and FBO, whose C-terminal contains other known domains or more than one known domain. Most of the previous studies upon F-box proteins belong to the FBL, FBK, and FBT subfamilies. These subfamily members participate in various biological processes: the FBL subfamily members mainly participate in phytohormone signaling, for instance, TRANSPORT INHIBITOR RESPONSE1 (TIR1) involved in auxin signaling [28] and CORONA-TINE INSENSITIVE1 (COI1) involved in jasmonic acid signaling [29]; the FBT subfamily members are involved in ethylene-dependent fruit ripening, such as TPL1/TPL [30], and the FBK subfamily members are involved in light signal regulation and phenylpropanoid metabolism, such as AFR/ZTL/FKF1/LKP [31,32] and KFB01/KFB20/KFB50/SAGL1 [33,34].

F-box proteins form SCF complexes through their N-terminal F-box domain and Skp1, Cullin1, and Rbx1 to degrade the C-terminal binding target proteins by the UPP. The UPP can be divided into four steps [35–37] (Figure 2b): (a) The ubiquitin-activating enzyme, E1, activates ubiquitin to form a thiol-ester linkage between its internal cysteine residue and C-terminal Gly residues of Ub in the presence of ATP. (b) The E1 transfer-activated ubiquitin binding to cysteine of the ubiquitin-conjugating enzyme, E2, to form new thioester. (c) Ubiquitin ligase, E3, catalyzes the formation of an isopeptide between the carboxyl at the C-terminal of Ub and the $\varepsilon$-amine of the substrate (in rare cases, the peptide is formed). (d) Finally, the 26S proteasome degrades the substrate into a polypeptide, after which monomeric ubiquitin is released. In the UPP, the ubiquitin chains formed by ubiqutination of substrates mainly consist of K48-linked polyUb [38]. Previous studies have found that branched K11/K48-linked polyUb not only participates in the UPP but also enhances affinity with proteasome subunit Rpn1 [39]. However, other ubiquitin chains (K6, K29, K63, etc.) formed by the specific Lys (K) sites could not participate in the UPP [40,41]. For instance, K63-linked polyUb is only involved in intracellular trafficking, DNA repair, and regulation of protein activity [42].

As mentioned above, the *F-box* gene family is the largest, with hundreds of members in plants, which indicates that F-box proteins could participate in the regulation of various physiological processes by binding to different target proteins. Studies have shown that F-box proteins are also involved in plant biotic and abiotic stress responses, in addition to regulating plant growth and development.

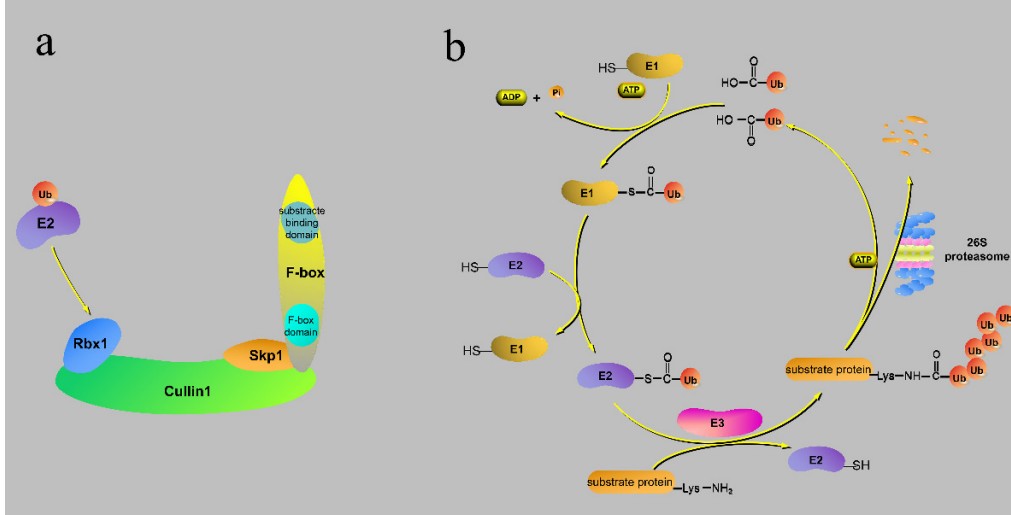

**Figure 2.** The SCF complex and the ubiquitin–proteasome pathway. (**a**) An F-box protein in the SCF complex. (**b**) The ubiquitin–proteasome pathway. E1 activates Ub in the presence of ATP and transfers Ub to E2. E2 then transfers Ub to the substrate-binding site of the E3 ubiquitin ligase. The 26S proteasome subsequently degrades the substrate. The RING-box (Rbx1), S-phase kinase-associated protein (Skp1).

## 3. The Function of F-Box Proteins in Plant

### 3.1. The F-Box Proteins and Plant Development

#### 3.1.1. Root Development

The root is one of the most important organs of plants. It can not only provide support for plants but also absorb nutrients and water. In addition, roots can perceive the change in external environment. It is difficult to observe roots that are underground, and many factors affect the root system architecture. Thus, there are some challenges in studying the root system. However, a large number of studies have shown that the *F-box* gene family is involved in root development. Carbonnel et al. found that the F-box protein MAX2 can inhibit the growth of primary roots and promote the growth of root hairs. This growth was possible by increasing the content of ethylene through the Karrikins signaling pathway [21]. Therefore, in the presence of Karrikins, MAX2 will degrade SMAX1 through the UPS and relieve the transcriptional inhibition of the *ACS7* gene, which is the rate-limiting enzyme for ethylene biosynthesis in *Lotus japonicus*. Moreover, Swarbreck et al. found that MAX2 can limit root skewing in Arabidopsis [43,44]. F-box genes can also regulate root development through auxin signals. Similarly, auxin promotes the expression of the F-box gene*CEGENDUO (CEG, AtSFL16)* in vascular tissue of primary roots, which inhibits lateral root formation [45]. In addition, Arabidopsis T-DNA insertion mutant and RNAi lines produce more lateral roots than WT, and the phenotype of the *ceg* mutant can be restored by complement. AUXIN UP-REGULATED F-BOX PROTEIN1 (AUF1) is an auxin-responsive F-box protein. Auxin induces *AUF1*, which can regulate root development by promoting auxin transport [46,47]. Under normal conditions, the roots of *auf1* were found to not be different from WT, and this phenotype was not different when indole acetic acid (IAA) was used. However, the root length of *auf1* was shorter than that of WT when NPA and TIBA were used. This result indicated that AUF1 can promote auxin transport. Furthermore, *auf1* was sensitive to cytokinin. After adding CK, the root of *auf1* was also shorter than that of WT. Therefore, Zheng et al. believed that AUF1 had an additional crosstalk between cytokinins and auxins [46]. Furthermore, some F-box proteins are not directly involved in plant hormone signaling control root development, such as *ARABIDILLO-1/-2* [48]. Compared with WT, the overexpression of *ARABIILLO-1/-2* lines produced more lateral roots, while the *arabidillo-1/-2* mutant produced fewer lateral roots.

### 3.1.2. Leaf and Stem Development

The leaf is the main place for photosynthesis, respiration, and evapotranspiration, which is crucial to the growth of plants. Leaf morphological structures are closely associated with crop yield and play an important role in response to stress. It was found that the F-box protein FBX92 can affect the leaf development of *Arabidopsis thaliana* by stimulating cell proliferation [49]. Compared with WT, transgenic plants overexpressing *AtFBX92* produce smaller leaves, and synthetic RNA silencing *AtFBX92* genes produce larger leaves. Interestingly, the Arabidopsis with heterologous gene of *ZmFBX92*, *AtFBX92* homologous gene, produces bigger leaves than WT. This is because the C-terminal of AtFBX92 protein has one F-box-associated domain more than that of the ZmFBX92 protein. Furthermore, the F-box gene *LEAF INCLINATION 4* (*LC4*) in rice can control the leaf angle of rice [50]. Compared with WT, overexpression *LC4* plants increased leaf angle by increasing the expansion and elongation of adaxial parenchyma cells, while the opposite was observed in LC4-Cas9 plants. In addition, LC4 can be induced by IAA but did not interact with the components of the IAA signal pathway. Similarly, the *HAWAIIAN SKIRT (HWS)* gene plays a certain function in leaf development [51]. In Arabidopsis, *HWS* deficiency leads to larger leaves, but in tomato, the *hws* mutant leaves become smaller, and lobule fusion occurs [52].

Leaf development ends in leaf senescence. A study on the *F-box* gene in leaf senescence has also been reported. In screening the senescence mutants of *Arabidopsis thaliana*, it was found that the *mineralara9* mutant had a delayed senescence phenotype [53]. It was then found in a later study that ORESARA9 (ORE9) was an F-box protein that can regulate leaf senescence by the UPP [54]. ORE9 is regulated by mitogen-activated protein kinase 6 (MPK6). MPK6 can promote the cleavage and nuclear translocation of C-terminal end of ORE3/EIN2 to stabilize EIN3 in the nucleus. This regulation leads to the binding of EIN3 to the *TTCAAA* element in the *ORE9* promoter, thus promoting the expression of *ORE9* [55]. Similarly, OsFBK12 can degrade OsSAMS1 by the SCF complex formed with Skp1, affecting ethylene synthesis and histone methylation [23]. Compared with the wild type, the ethylene content of OsFBK12-RNAi and *OsSAMS1* overexpression lines increased, resulting in aging. In contrast, overexpression of *OsFBK12* and OsSAMS1-RNAi decreased ethylene content, thus delaying senescence.

The stem is one of the six major organs of plants, which can transport water and inorganic salts absorbed by roots, as well as the products of leaf photosynthesis to other organs of plants. Research findings showed that some *F-box* genes can regulate stem development, in addition to regulating leaf development. Liu et al. found that after silencing *SlGID2*, an F-box gene, in tomato, the leaf color turned darker, the size became smaller, and the stem cells became smaller and more compact, resulting in dwarfing [56]. However, this dwarfing phenomenon was not caused by the lack of GA in vivo and could not be rescued by exogenous GA. In contrast, the GA content in *SlGID2i* was higher than that in WT. Similarly, Wang et al. found that the Arabidopsis F-box protein SAP can ubiquitinate PPD proteins to promote the proliferation of meristem cells, which regulates organ size [57]. The *sap* T-DNA insertion mutant (early termination) therefore generates smaller leaves and shorter plant height than WT. Moreover, *SMALL LEAF AND BUSHY1 (SBL1)*, *SAP* homologous genes, can regulate leaf size and the number of lateral branches by altering cell proliferation [58]. Leaves of *sbl1* are thus smaller than those of WT, but an increase in the number of lateral branches is observed.

### 3.1.3. Flower and Fruit Development

*UNUSUAL FLORAL ORGANS* (*UFO*) was the first identified F-box gene in plants, one that can regulate floral organ development [59]. Flower development is mainly controlled by *ABC* genes, wherein *A* genes control the growth of sepals and petals, *B* genes control the growth of petals and stamens, and *C* genes control the growth of stamens and carpels. The AtUFO can mediate the protein hydrolysis of inhibitors of *APETALA3 (AP3)* and *PISTIL-LATA (PI)*, class B functional genes, by interacting with *AtASK* to promote the expression of *AP3* and *PI* [60]. In Arabidopsis, the *Atufo* mutation causes a series of floral abnormalities,

such as in the number of sepals, flower size, and so on [59]. Similarly, the *ufo* mutation in Cymose *Aquilegia coerulea* results in small flowers, increased sepals, and decreased petals and stamens [61]. In rice, the functions of class B genes*OsMADS4* and *OsMADS16* are largely conservative, being regulated by the F-box gene*DDF1* [62]. Previous studies have also shown that the palea and lemma of *ddf1-1* mutant spikelets were deformed, resulting in untight closure aberrant pistil-like organs. The *OsMADS4* and *OsMADS16* genes were also significantly downregulated, whereas the *DL* gene was significantly upregulated in *ddf1-1*. This result indicates that *DDF1* can promote *OsMADS4* and *OsMADS16* gene expression and inhibit *DL* gene expression. Similarly, the photo-responsive F-box protein*FOF2* adjusts flowering time by promoting the expression of the *FLC* gene [63]. Overexpression of *FOF2* in Arabidopsis can delay flowering, while the dominant-negative mutant results in early flowering. In addition, *FOF2* transgenic plants retained a photoperiod response. Compared with long-day conditions, overexpression of *FOF2* bloomed late under short-day conditions. *HWS* can regulate flower and fruit development in addition to the above-mentioned function that controls leaf development [51]. The sepals of *Arabidopsis thaliana* mutant *hws* showed fusion, while anther filaments were fused to the side of the silique. Moreover, the siliques of *hws* were bigger than WT due to aberrant septum development. Later, it was found that *HWS* controlled flower size and floral organ number by regulating the expression of *CUC1* and *CUC2* genes [64]. Furthermore, the tomato *hws* mutants not only cause abnormal flower development but also promote parthenocarpy [52].

## 4. The F-Box Proteins and Biotic Stress

Biotic stress has adverse effects on plant growth and development. In the process of plant growth, pathogens and pests attack plants. As a result, plants have developed many defense mechanisms to protect themselves during the process of evolution. Stomatal closure is an important part of plant resistance to bacterial innate immune response. When pathogens come into contact with leaves, plants resist the pathogens by closing their stomata [65]. In addition to affecting plant growth and development, MAX2 is a protein that is essential for resisting pathogen invasion in plants. Previous studies have shown that the stomatal closure function of Arabidopsis mutant *max2* was impaired, and stomatal conductance was enhanced, which allows more pathogens to enter the plant plastids, leading to enhanced susceptibility [66]. Furthermore, the F-box-Nictaba protein is a plant lectin that plays an important role in plant defense [67,68]. *Pseudomonas syringae* can also induce the expression of *F-box-Nictaba* gene in *Arabidopsis thaliana*. Compared with WT and *F-box-Nictaba* knock-out plants, overexpression of *F-box-Nictaba* plants showed relatively slight disease symptoms after infection with *Pst DC3000*. *OsDRF1* also increases plant tolerance to *Pseudomonas syringae*. It can also increase tolerance to viruses [69]. The heterologous expression of *OsDRF1* in tobacco increased the expression of defense genes; for instance, *PR-1a* and *Sar8.2b* in vivo increases the resistance of plants to cauliflower virus after tomato cauliflower virus infection. Similarly, *Verticillium dahlia* can induce the expression of *GhACIF1* gene, which can interact with SKP1 to form the SCF complex to mediate the PevD1-HR/SAR pathway [25]. Overexpression of *GhACIF1* enhances resistance to *Verticillium dahliae* in Arabidopsis, while silencing of *GhACIF1* confers sensitivity to *Verticillium dahliae* in cotton. VpEIFP1 can also form SCF complexes with SKP1 and CUL1 to mediate the degradation of Trxz, thus inducing ROS accumulation and activating defense reactions, thereby inhibiting the growth and development of PM [70]. The heterologous expression of *VpEIFP1* in Arabidopsis accelerated the accumulation of $H_2O_2$ in vivo as well, as well as upregulating the expression of *ICS2*, *NPR1*, and *HSP* genes, resulting in plant tolerance to powdery mildew.

## 5. The F-Box Proteins and Abiotic Stress

Abiotic stress is also one of the key factors affecting plant growth and development. Abiotic stress mainly includes water stress (drought, flood), salt stress (salt, alkali), temperature stress (high temperature, low temperature), and toxic metal stress [71–73]. As

sessile organisms, plants adapt to the environment only by a self-regulation system under abiotic stress conditions. Previous studies have shown that the F-box gene family members can participate in various abiotic stress responses. Drought stress adversely affects plant growth and development and is the main reason for crop failure [73]. Facing drought stress, plants reduce the damage caused by drought stress by regulating stomatal conductance and decreasing the transpiration rate to reduce moisture loss. Li et al. also found that the F-box gene *SITLFP8*, induced by osmotic stress in tomato, can change the stomatal density of leaves by affecting endoduplication and cell size [74]. Furthermore, overexpression of *SILFP8* enhances drought resistance of tomato by reducing stomatal density and decreasing the transpiration rate to improve water use efficiency. In contrast, SITLFP8-Cas9 plants increased stomatal density and transpiration rate, showing sensitivity to drought. Similarly, MdMAX2 enhances drought resistance of plants by promoting stomatal closure to reduce water loss [75]. Overexpression of *TaFBA1* in Arabidopsis enhances drought resistance by decreasing the content of $H_2O_2$, $O_2^-$, and MDA in vivo. However, interestingly, the transpiration rate and stomatal conductance of transgenic plants increased significantly. This increase in rate is proposed to be because increased stomatal opening can enhance $CO_2$ absorption to maintain high-carbon fixation in leaves, further reducing ROS accumulation [76,77]. In addition, *TaFBA1* can participate in the regulation of salt stress, oxidative stress, and high-temperature stress [78–80]. Compared with the wild type, the germination rate, root length, chlorophyll content, the net photosynthetic rate, and antioxidant enzyme activity of *TaFBA1*-overexpressing tobacco was significantly increased under oxidative and salt stress conditions.

Aluminum is the most abundant metal element in the crust. Under acidic conditions, some trivalent aluminum ion was dissolved to inhibit root growth of plants, thereby affecting the growth of plants. Acid soils account for more than 30% of arable land in the world, and therefore aluminum toxicity is also considered the second largest abiotic stress after drought [81,82]. F-box protein RAE1 can change plant tolerance to aluminum stress by regulating the stability of the transcription factor STOP1 [24]. The STOP1 transcription factor is a key regulator of the malic acid transporter *AtALMT1* (a critical Al-tolerance gene). Mutation in STOP1 greatly suppress the expression of *AtALMT1* gene [83]. RAE1 ubiquitinates STOP1 by the ubiquitin-26S proteasome pathway. STOP1 can positively regulate the expression of *RAE1* by binding to the promoter region of *RAE1*. Thus, a negative feedback loop was formed. However, the ubiquitination of STOP1 mediated by *RAE1* was inhibited under Al stress. This inhibition is proposed to be because of the post-translational modification of STOP1 caused by Al stress.

In addition to the above two abiotic stresses, salt and low-temperature stresses are also main factors limiting plant growth. The F-box protein EST1 is a negative regulator of salt stress. It can interact with Skp1 to form an SCF complex for MKK4 ubiquitination degradation, which decreases the activity of MKK4-MPK6 cascade reactions, decreasing in $Na^+/H^+$ antiporter activity [80]. The $Na^+/H^+$ antiporter activity of the loss-of-function mutant was significantly increased, which reduced $Na^+$ accumulation in vivo, resulting in more tolerance to salt. In contrast, overexpression of *EST1* plants was more sensitive to salt stress than WT. The F-box gene *OsMsr9* in rice was induced by various stresses, and it can increase the salt tolerance of plants [84]. Under salt stress, the root length and plant height of *OsMsr9*-overexpressed plants were significantly higher than those of WT, and the content of proline and soluble sugars were improved in vivo. Similarly, the F-box protein LTSF1/2 in pepper is the key subunit of the SCF complex, which can regulate tolerance to low-temperature stress by activating antioxidant enzyme activity [85]. Under low-temperature stress, the growth of transgenic tobacco with heterologous expression of *LTSF1* was found to be significantly higher than that of WT, and the expression of antioxidant enzyme genes such as *GST* and *CAT* in vivo was also significantly increased. In addition, the inhibition of *CaF-box* (GenBank, JX402925) gene by VIGS (virus-induced gene silencing) decreased the expression levels of the cold-induced genes (*ERD15* and

*KIN1*) [86]. In summary, these results suggest that the CaF-box affects cold tolerance in plant by regulating the expression of *ERD15* and *KIN1*.

## 6. F-Box Proteins and Phytohormone Signaling

Phytohormone is a small molecule compound in plants that acts as a chemical messenger and affects the physiological function of plants at low concentrations. The complex signal network of phytohormones can make plants effectively resist external stress and balance growth. Thus, it plays an important role in regulating plant growth and development and responding to external stress [87]. Signal transduction pathways of plant hormones generally include positive and negative regulators, hormone receptors, and downstream response genes. These pathways include three strategies, namely, protein degradation control, protein phosphorylation, and their combination [88]. The signal transduction of protein degradation control mainly includes auxin, jasmonic acid (JA), gibberellic acid, strigolactones, and karrikin signaling. The signal transduction pathways of protein phosphorylation control mainly include cytokinin and brassinosteroids. However, the ethylene signaling pathway involves the combination of two strategies. The signal pathway of protein degradation can activate the expression of hormone responsive genes through the SCF complex, ubiquitinating its target and releasing downstream transcription factors. On the other hand, the signal pathway of protein phosphorylation control regulates the expression of target genes through triggering a reversible phosphorylation cascade reaction of receptor kinases located on the membrane by phytohormone.

### 6.1. Auxin Signaling

Auxin is a plant hormone that regulates the growth and development of plants. It can transmit information over long and short distances. Natural auxins include indole-3-acetic acid (IAA), indole-3-butyric acid (IBA), 4-chloroindole-3-acetic acid, and phenylacetic acid [89–91]. Among them, IAA is the richest auxin in plants, and the only endogenous molecule that directly activates auxin signals. The core of auxin signal transduction pathway mainly occurs in the nucleus, which activates the transcription of the plant auxin response gene through the SCF$^{TIR/AFB}$-proteasome pathway [90]. This transduction pathway includes the auxin receptor protein TRANSPORT INHIBITOR RESPONSE1/AUXIN SIGNALING F-BOX PROTEIN (TIR1/AFB), the transcription factor AUXIN RESPONSE FACTOR (ARF), the transcriptional repressor AUXIN/INDOLE ACETIC ACID (AUX), and auxin. Here, IAA acts as a molecular glue to promote the binding of AUX/IAA with TIR/AFB [91,92]. The AUX/IAA protein can inhibit ARF, a transcription factor that responds to auxin [93]. TIR1/AFB, an F-box protein, can ubiquitination degrade its target protein, AUX/IAA, by forming SCF complexes, and its transcription level is regulated by *miR393* [28,94]. In plants, when auxin levels are low or no auxin exists, the complex, composed of a transcriptional repressor AUX/IAA and its co-inhibitor TOPLESS (TPL), will inhibit the auxin response gene expression (Figure 2a). When the auxin level is increased, IAA enhances the recognition and binding of TIR/ARF and AUX/IAA. Then, the AUX/IAA protein is degraded by SCF$^{TIR/AFB}$ through ubiquitination, which leads to the disinhibition of ARF (Figure 3a).

### 6.2. JA Signaling

JA is an endogenous growth regulator found in higher plants. This phytohormone can promote plants to respond to external damage (mechanical, insect damage) and pathogen infection, in addition to participating in plant growth and development [95,96]. In plants, free jasmonic acid can generate JA-Ile, a class of active substances of JA, through the catalysis of jasmonate amide synthase [97,98]. JA-Ile can activate JA signal transduction. The JA signal pathway is similar to auxin pathway. Here, the F-box protein CORONATINE INSENSITIVE1(COI1) acts as the receptor of JA-Ile, and JA-Ile acts as a molecular glue-like IAA [99]. JA-Ile can promote the binding of COI1 and JAZ to form COI1–JA-Ile–JAZ complex [29,100]. JAZ is a transcriptional repression protein that can form JAZ–NINJA–TPL

ternary complex through novel interactor of JAZ (NINJA) protein and TPL, in addition to binding to some transcription factors through ZIM domain, such as the MYC transcription factor family [101,102]. External environmental stress can induce JA-Ile [103], promoting the binding of COI1 and JAZ. Then, JAZ was ubiquitinated by SCF$^{COI1}$, and transcription factor activity was restored (Figure 3b).

### 6.3. Gibberellic Acid Signaling

Gibberellin, an essential phytohormone in plant growth, plays an important role in the first green revolution. Many forms of GA exist in plants, and the main active forms are GA1, GA3, GA4, and GA7 [104,105]. Research has found that GAs play an important part in seed germination, stem elongation, and flower development [106–108]. In the first green revolution, GA-insensitive dwarf mutants were used to improve the lodging resistance capability, thus greatly increasing crop yield. GA-insensitive dwarf mutants are commonly due to deletion or mutation of the N-terminal of the DELLA protein, which makes it unable to be degraded [104,109,110]. The GA signal pathway of GA-induced protein degradation mainly includes the GA receptor GIBBERELLIN INSENSITIVE DWARF1 (GID1), the F-box protein GIBBERELLIN INSENSITIVE DWARF2/SLEEPY1 (GID2/SLY), the transcriptional repressor DELLLA protein, and the transcription factor PHYTOCHROME-INTERACTING FACTOR 1 (PIF1). The GA signal transduction mechanism is similar to auxin and JA. The difference is that the F-box protein GID2/SLY does not act as the receptor of phytohormones, and is only responsible for recruiting DELLA proteins for ubiquitination degradation [47,91]. DELLA proteins belong to the large GRAS family member, which can interact with various transcription factors and changes its activity [111,112]. When GID1 perceives and binds to GA, its conformation will change. The N-terminal of GID1 then wrapped GA and exposed a removable lid, which could interact with DELLA protein [113,114]. Subsequently, the complex of GA–GID1–DELLA bind to F-box protein GID2/SLY, and then the DELLA protein was degraded by ubiquitination, which leads to the disinhibition of DELLA and activates GA response genes (Figure 3c).

### 6.4. Strigolactones and Karrikin Signaling

Strigolactones are newly identified plant hormones that are derived from β-carotene. SLs can not only regulate axillary bud growth [115,116], leaf senescence [117], and root development [21], but can also respond to multiple stresses [118] and interactions with plant symbiotic bacteria [119]. In plants, SL is perceived by the α/β hydrolase DWARF14 (D14). D14 is a bifunctional receptor that can also hydrolyze SL, in addition to sensing SL [120]. When D14 binds SL, D14 can cut the ABC and D rings of SL through a nucleophilic attack. Then, the ABC-FTL is released, and the HMB maintains a covalent connection with D14, thus promoting the conformational change of D14. This conformational change promotes the interaction between D14 and the F-box protein MAX2/DWARF3 (D3). Furthermore, SL can promote the interaction between D14 and SMAX1-like (SMXL) family proteins, such as SMXL6, SMXL7, and SMXL8 [91,121]. In the absence of SL, D53/SMXL can bind to the transcription corepressor TPL to inhibit the expression of downstream genes (Figure 3d). However, when SL is present, SL can promote the binding of D14 to MAX2 and SMXL6/7/8 proteins, resulting in the ubiquitination of D53/SMXL6/7/8 by SCF$^{MAX2/D3}$, which releases the transcriptional inhibition of downstream genes [122–124].

It was found that MAX2 not only degraded SMXL6/7/8 but also degraded SMXL1. The SMXL1 protein is also involved in the signal transduction pathways of the karrikins (KARs) [125]. KARs are a class of butenolides found in wildfire smoke that can promote seed germination. Its structure is similar to SL, which results in KAR signal pathway similar to SL (Figure 3d), but being independent of each other [126,127]. KARs also have their own individual receptor, karrikin insensitive 2 (KIA2), which is an ancient paralog of D14 [128]. KAI2 is also a bifunctional receptor like D14, but its binding pocket is smaller than that of D14, which results in KAI2 only binding KARs, not SL [129].

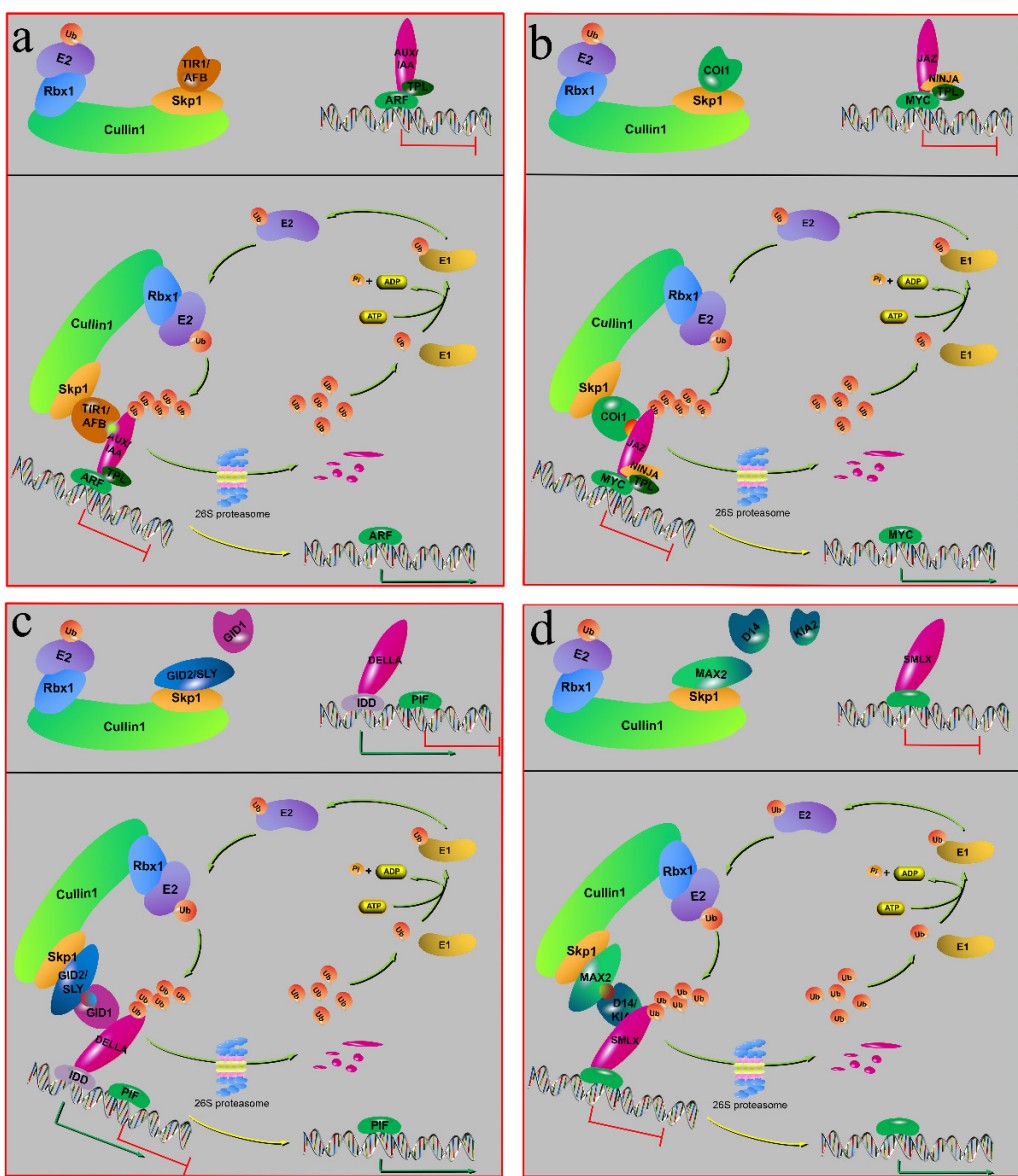

**Figure 3.** The plant hormone signaling pathways involved in F-box proteins. The depiction of the mode of five phytohormones' signal transduction involved in the protein degradation control. (**a**) Auxin (indole-3-acetic acid, IAA) signaling pathway. (**b**) Jasmonic acid (JA) signaling pathway. (**c**) Gibberellic acid (GA3) signaling pathway. (**d**) Strigolactone (SL) and karrikin (KAR1) signaling pathways. D14 is the receptor of strigolactone; KIA2 is the receptor of karrikin. The upper layer is without hormones or with few hormones, while the lower layer shows the presence of hormones.

*6.5. Ethylene Signaling*

Ethylene is a gaseous phytohormone that can regulate the growth and development of plants [130]. In terms of growth, ethylene can also regulate cell size by limiting cell elongation. In terms of development, ethylene can promote plant growth, fruit ripening, and organ senescence [131–133]. At present, the ethylene signal transduction pathway has been deeply understood, which involves the combination of the two strategies of protein degradation and protein phosphorylation control. The main components of the ethylene signal transduction pathway include ethylene receptors such as ETHYLENE RESPONSE 1 (ETR1), ETHYLENE RESPONSE SENSOR 1 (ERS1), ETHYLENE RESPONSE 2 (ETR2), ETHYLENE-INSENSITIVE 4 (EIN4), and ETHYLENE RESPONSE SENSOR 2 (ERS2); CTR1 protein kinase; EIN2, A PROTEIN OF Nramp-like protein in natural resistant macrophages; transcription factors such as ETHYLENE-INSENSITIVE 3(EIN3), EIN3-LIKE1(EIL1), and

EIN3-LIKE2(EIL2); and ethylene response genes *ERF* [131,134,135]. Among them, the F-box proteins regulate the stability of transcription factors through the UPP. In the absence of ethylene, the ethylene receptor ETHYLENE RESPONSE1 (ETR1) located in the endoplasmic reticulum is activated by CTR1 and promotes the phosphorylation of EIN2, which results in that EIN2 being recognized and degraded by SCF$^{ETP1}$ formed by the F-box protein EIN2 TARGETING PROTEIN1 (ETP1) (Figure 4). In the presence of ethylene, ETR1 can bind with ethylene, leading to CTR1 inactivation. At this time, EIN2 will not be degraded, but its C-terminal will be cut off and translocated into the nucleus, which inhibits the ubiquitination degradation of EIN3 by SCF$^{EBF}$ and promotes the expression of *ERF* and ethylene-responsive genes [135–137].

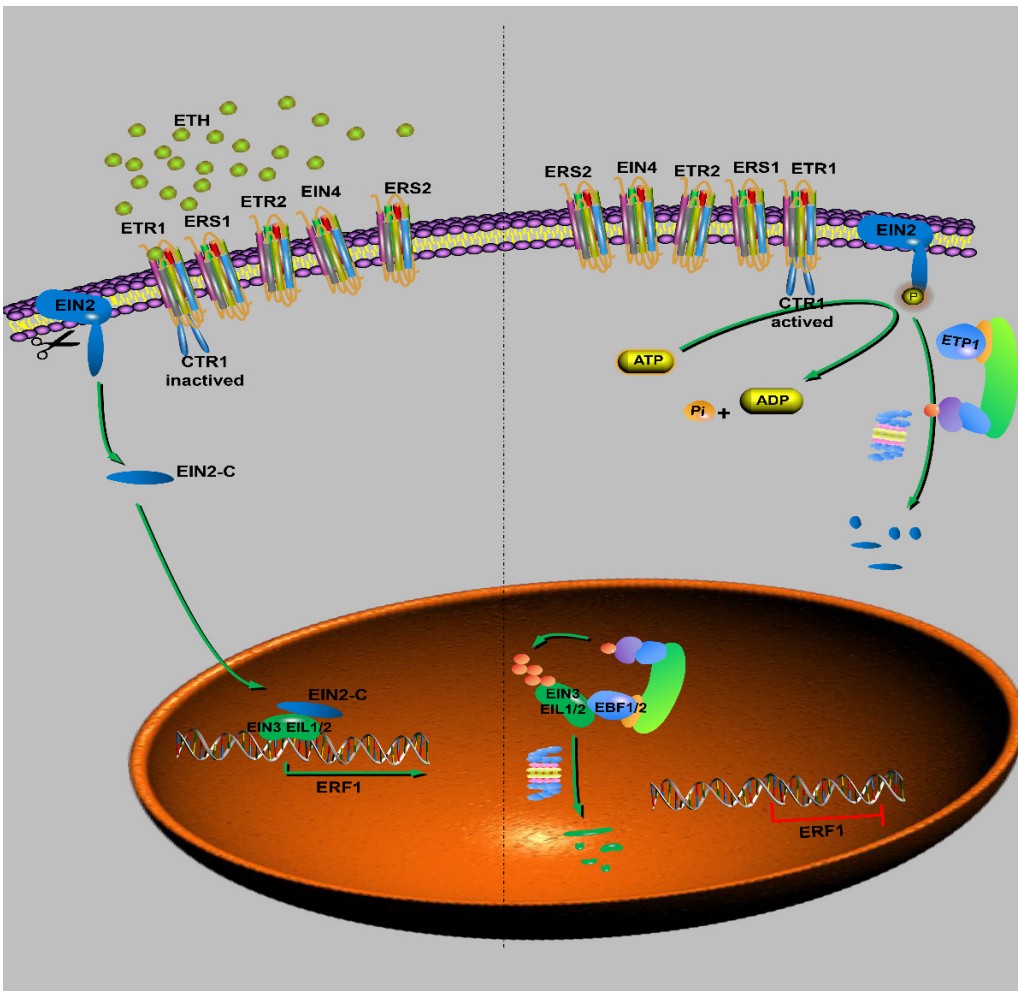

**Figure 4.** The ethylene signaling pathway involved in F-box proteins. The left side of the virtual line shows the presence of ethylene; the right is represented by absence of ethylene or with less ethylene content.

### 6.6. Other Phytohormone Signaling

In addition to participating in the above phytohormone signaling, F-box proteins play an important role in salicylic acid (SA) signaling and cytokinin (CK) signaling. As is known, SA is an important endogenous phytohormone for plants to resist external stimuli [138]. CPR1, a F-box protein, can reduce plant resistance to external stimuli by the regulation of SA content and target ubiquitin degradation of the suppressor of npr1-1, constitutive 1 (SNC1) in plants [139,140]. Moreover, SA-induced SNC1 can promote the expression of resistance genes EDS1/PAD4 and enhance plant resistance to external stimuli [141]. Furthermore, a feedback loop exists between *SNC1* and *DND1/2* in that SNC1-TPL as a repressor can

inhibit *DND1/2*, and DND1/2 can also inhibit the expression of *SNC1* [142]. Moreover, Mo et al. [143] proved that *SlFBX5, SlFBX24, SlFBX33, SlFBX38, SlFBX42, SlFBX51, SlFBX65, SlFBX67,* and *SlFBX79* belonging to the F-box gene family showed a downregulated trend after 48h of SA treatment. Furthermore, several studies showed that F-box protein (D3) binding to D14 in the presence of SL can promote ubiquitin degradation of D53 by SCF$^{D3}$, which can increase the expression of *OsCKX9* to reduce the CK content [115,144]. In summary, F-box proteins extensively participate in phytohormone signaling.

## 7. Future Issues

The F-box protein plays an important role in plant growth and development as well as resistance to external environmental stimuli. Current studies have shown that the F-box protein mainly forms SCF complexes through the interaction of the F-box domain and Skp1 or Skp1-like proteins to ubiquitinate and degrade its target protein. Whether other proteins that can bind to F-box domain is still a question to be explored. Moreover, although the functions of some *F-box* genes have been identified and great progress has been made in the phytohormone signaling pathway involved in F-box protein, for the entire *F-box* gene family, our current understanding of F-box is a drop in the ocean. Therefore, we should be able to find the function of more uncharacterized F-box proteins and the proteins interacting with them. These future studies will thus clarify the various physiological functions and biochemical mechanisms of F-box proteins in plants. Due to the large gene family, there are many homologous genes with redundant functions, which brings difficulty in studying the function of *F-box* genes. However, this should not be of concern; with the development of technology, we will identify the function of a large number of *F-box* genes by effective techniques soon, such as dominant inactivation mutants, a combination of classical genetics and modern gene technology, which is an effective way to study these genes. In addition, future studies will focus more on the study of pathways involved in F-box proteins, thus making it a popular field of future research.

**Author Contributions:** Writing—original draft preparation, K.X.; writing—review and editing, H.L. and Y.Z.; visualization, N.W. and W.Y.; funding acquisition, X.L. All authors have read and agreed to the published version of the manuscript.

**Funding:** This work was supported by the Natural Science Foundation of Science Technology Department of Jilin Province (20190201259JC), the National Natural Science Fundation of China (32171937, 32001464), the Scientific Research Foundation of Hainan University Program (Y3AZ20024), and the Hainan Provincial Natural Science Foundation of China (321QN182).

**Institutional Review Board Statement:** Not applicable.

**Informed Consent Statement:** Not applicable.

**Data Availability Statement:** All data included in this study are available upon request by contact with the corresponding author.

**Conflicts of Interest:** The authors report no declarations of interest.

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
