# Peer review of "The Biological Function and Roles in Phytohormone Signaling of the F-Box Protein in Plants"

_agronomy, doi:10.3390/agronomy11112360_

Round 1

Reviewer 1 Report

The paper presents a literature revision on F-box E3 ligases that include a description of number of F-box, major subfamilies and some biological roles associated to. While the subject is suitable for Agronomy, the revision should be needing an increasing link between sub-subjects through the paper. I think, the manuscript does not display a connection between subfamilies described in the first section, structure, and role of F-box ligases. The paper needs to improve the following comments:

1.- Line 95, The E2 enzymes are also important on SCF-type E3 ligase and is missing in the passage and in the Fig 2a.

2.-Lines 103, The variety of Ub-linkage is one of the big challenges in UPP and should be useful to add a brief discussion.

3.- Line 124, The reference has a misspelling error.

4.- Double check the manuscript for typo errors. For instance, involve (line 312), include (Line 420), etc.

5.- Line 402, The Fig 3e is missing in the manuscript.

6.- It should be useful to show more connection between subfamilies and the plant role through the manuscript or add a brief discussion.

Reviewer 2 Report

The manuscript entitled 'The biological function and roles in phytohormone signaling of the F-box protein in plants' describes the classification, function, and involvement in the hormone signaling pathways. Though the writing is well described, there is a lack of description about SA signaling and the adaptor proteins involved in SA -induced immunity which is similar to F-Box proteins. For instance, the authors did not mention the effects of SA on SlFBX genes. Similarly, CPR1, which negatively regulates both SA-dependent and SA_independent signaling. The authors should also mention about CAF box involved in abiotic stress. 

The figures need to be improved. The writings are not visible. I suggest the authors increase the font of each figure as well as the resolution.

Line 452. We should be able to find...
